# Intranasal Cerium Oxide Nanoparticles Ameliorate Cognitive Function in Rats with Alzheimer’s via Anti-Oxidative Pathway

**DOI:** 10.3390/pharmaceutics14040756

**Published:** 2022-03-30

**Authors:** Syed Mohammad Danish, Anshul Gupta, Urooj Ahmad Khan, Nazeer Hasan, Farhan Jalees Ahmad, Musarrat Husain Warsi, Ahmed M. Abdelhaleem Ali, Ameeduzzafar Zafar, Gaurav Kumar Jain

**Affiliations:** 1Department of Pharmaceutics, School of Pharmaceutical Education and Research, Jamia Hamdard, New Delhi 110062, India; drsyed249@gmail.com (S.M.D.); nazeerhasan1994@gmail.com (N.H.); farhanja_2000@yahoo.com (F.J.A.); 2Department of Pharmaceutics, Delhi Pharmaceutical Sciences and Research University, New Delhi 110017, India; 3Department of Pharmaceutics, School of Medical and Allied Sciences, KR Mangalam University, Gurgaon 122103, India; urooj_z14@yahoo.co.in; 4Department of Pharmaceutics and Industrial Pharmacy, College of Pharmacy, Taif University, P.O. Box 11099, Taif 21944, Saudi Arabia; a.mali@tu.edu.sa; 5Department of Pharmaceutics, College of Pharmacy, Jouf University, Sakaka 72341, Al-Jouf, Saudi Arabia; azafar@ju.edu.sa; 6Center for Advanced Formulation Technology, Delhi Pharmaceutical Sciences and Research University, New Delhi 110017, India

**Keywords:** biochemical estimation, morris water maze test, nose to brain, passive avoidance test, rivastigmine

## Abstract

Cerium oxide nanoparticles (CNPs), owing to their antioxidant property, have recently emerged as therapeutic candidate for Alzheimer’s disease (AD). However, intravenous CNPs are limited due to their poor physicochemical properties, rapid blood clearance and poor blood–brain penetration. Thus, we developed intranasal CNPs and evaluated its potential in experimental AD. CNPs were synthesized using homogenous precipitation method and optimized through Box–Behnken Design. The formation of CNPs was confirmed by UV spectroscopy and FTIR. The optimized CNP were spherical, small (134.0 ± 3.35 nm), uniform (PDI, 0.158 ± 0.0019) and stable (ZP, −21.8 ± 4.94 mV). The presence of Ce in CNPs was confirmed by energy-dispersive X-ray analysis. Further, the X-ray diffraction spectra revealed that the CNPs were nano-crystalline. The DPPH assay showed that at concentration of 50 µg/mL, the percentage radical scavenging was 95.40 ± 0.006%. Results of the in vivo behavioral studies in the scopolamine-induced Alzheimer rat model showed that intranasal CNPs dose dependently reversed cognitive ability. At dose of 6 mg/kg the morris water maze results (escape latency, path length and dwell time) and passive avoidance results (retention latency) were significantly different from untreated group but not significantly different from positive control group (rivastigmine patch, 13.3 mg/24 h). Further, biochemical estimation showed that intranasal CNP upregulated the levels of SOD and GSH in brain. In conclusion, intranasal CNPs, through its antioxidant effect, could be a prospective therapeutics for the treatment of cognitive impairment in AD.

## 1. Introduction

Alzheimer’s Disease (AD) is a progressive neurodegenerative disorder characterized by behavioral impairment and cognitive dysfunction that markedly interferes with occupational and social functioning [1]. A remarkable amount of evidence has demonstarted that reactive oxygen species (ROS) and oxidative stress is inextricably linked with several neurodegenerative disorders including AD. Brain, due to high oxygen demand and due to presence of abundant peroxidation-susceptible lipids, proteins and nucleic acid is highly vulnerable to ROS imbalance [1,2,3]. ROS like hydroxyl radical (•OH), superoxide anion (O2 •−), singlet oxygen (^1^O_2_), and non-free radical species such as hydrogen peroxide (H_2_O_2_) are responsible for cellular oxidative damage [4,5]. The accumulation of oxidized products (plaques) seriously alters the cellular functions and deposition in the deeper brain tissues such as hippocampus and cerebral cortex leads to loss of memory and thinking and impairment of cognitive and behavioral functions, characterstics of AD [4,5].

Currently, AD is managed by use of cholinesterase inhibitors (donepezil, rivastigmine) and NMDA receptor inhibitors (memantine) [2,3]. However, several antioxidant molecules such as vitamins, curcumin, berberine, quercetin, resveratrol and melatonin had shown promising results in the management of AD [6]. Owing to their unique antioxidant property, cerium oxide nanoparticles (CNPs) have recently emerged as potential therapeutics for AD [7,8,9,10]. In CNPs, cerous (Ce^+3^) and ceric (Ce^+4^) co-exist in the core while oxygen envelops the core. The Ce (III) scavenges ROS and is oxidized to Ce (IV) and Ce (IV), then slowly reduced back to Ce(III). Thus, in contrast to elemental cerium, this unique property conferred by the nano-sized particle allows cerium to cycle between these two states creating regular oxygen vacancy in the crystal lattice to ensure continuous oxygen scavenging [11]. Furthermore, CNPs mimics superoxide dismutase (SOD), catalase, peroxidase, and oxidase enzymes which are known to decrease ROS levels by converting H_2_O_2_ into oxygen, water, etc [12]. The previous reports describe the therapeutic potential of CNPs in AD either using in vitro neuronal cell lines [10] or following intravenous CNPs in animal model [7,8,9,10]. However, clinical translation of intravenous CNP for the management of AD is limited due to its poor physico-chemical properties, rapid blood clearance and poor blood brain penetration.

The importance of intranasal (i.n.) route to bypasses the blood-brain barrier and to deliver the drugs directly to the brain is well established and using this route, researchers are able to reverse neurodegeneration [13,14,15,16,17]. Intranasal delivery not only eliminates the need for invasive systemic administration but also reduces the potential side effects particularly when localized brain delivery is required [13,14,15,16,17]. This is likely because of the inimitable connections between the nasal mucosa and the olfactory cells and trigeminal nerves [14]. Both drug and nanoformulations could be targeted to the olfactory region and memory areas following i.n. administration. Further, report also showed that nano-sized particles reach directly into CSF by absorption through the nasal epithelium into subarachnoid space [18]. Thus, in the present study, we evaluated the effect of intranasal CNPs on the cognitive capability of scopolamine-induced AD rats, as modeled through the Morris water maze (MWM) and a passive avoidance (PSA) test. To our best knowledge this paper, for the very first time, reports the dose and prospective use of intranasal CNPs in AD.

## 2. Materials and Methods

### 2.1. Materials

Cerium nitrate hexahydrate (Ce(NO_3_)_3_.6H_2_O, 99.9%), ammonia and scopolamine were procured from Sigma Aldrich, St. Louis, MO, USA. Analytical grade methanol and ethanol were purchased from Merck, Germany and acetonitrile was purchased from Thomas Baker Chemicals, Mumbai, India.

### 2.2. Development of CNPs

#### 2.2.1. Synthesis of CNPs

CNPs were synthesized by the modification of homogenous precipitation method [19]. Briefly, cerium (III) nitrate hexahydrate (0.04 M) was dissolved in a 20 mL mixture of methanol and water (70:30 *v*/*v*). The ammonia (3 M, 20 mL) was then added to the mixture maintained at 50 °C and the reaction was allowed to proceed for 2 h at 500 rpm with continuous stirring. A yellow suspension thus obtained was centrifuged (2000 rpm, 5 min), washed twice with methanol-water mixture to yield CNPs. The CNPs were dried overnight at 50 °C and stored at 4 °C until further use.

#### 2.2.2. Optimization of CNPs

The optimization of CNPs was carried using 3-factor, 3-level Box–Behnken design (Stat-Ease, Inc., version 8.0.7.1, Minneapolis, MN, USA). The effect of independent variables such as cerium nitrate concentration (0.01—0.04%), alcohol concentration (20—70%) and temperature (30—70 °C) on size and zeta potential were determined with 17 experimental runs (Table 1). The target was set at minimum particle size and maximum zeta potential. The nonlinear quadratic equation was:Y = b_0_ + b_1_ × _1_ + b_2_X_2_ + b_3_X_3_ + b_12_X_1_X_2_ + b_13_X_1_X_3_ + b_23_X_2_X_3_ + b_11_X_12_ + b_22_X_22_ + b_33_X_32_(1)
where, Y = response calculated for each factor level integration, b_0_ = intercept, b_1_ to b_3_ = regression coefficients, X_1_ to X_3_ = levels of independent variables, X_1_ X_2_ and X_i2_ (I = 1, 2 or 3) = portray interaction and quadratic terms respectively. deleted

#### 2.2.3. Characterization of CNPs

The light absorption study of CNPs was carried out by a double beam UV spectrophotometer (Shimadzu, Kyoto, Japan) in the range of 200–800 nm. Infra-red spectra of CNP was measured by the KBr pellet method using FTIR (Perkin Elmer, USA) in 400 to 4000 cm^−1^ wavelength range. The size and morphology of CNPs was determined by transmission electron microscope (TEM, 1200 EX, JEOL, Austin, TX, USA) and scanning electron microscope (SEM, JEOL, JSM6390LV, Austin, TX, USA). TEM samples were analyzed after mounting and drying a drop of CNP suspension on carbon-coated copper grids, whereas SEM analysis was conducted at 20 kV and a resolution of 3 nm. Size and zeta potential measurements were conducted at 90° using dynamic light scattering (Nano-ZS90, Malvern, Groovewood Road, Malvern, UK). Crystallanily of the CNPs was determined by X-ray Diffraction (XRD, X’Pert, Malvern, UK). X-ray intensity of detector system was 2ϴ in the range between 20° and 80° and a constant angle (ω = 1.0°) was maintained between the incident X-ray beam and CNP surface. The presence of cerium in the CNP was confirmed by energy-dispersive X-ray analysis (EDXA) performed on TEM (1200 EX, JEOL, Austin, TX, USA).

### 2.3. Antioxidant Activity of CNPs

The antioxidant activity of the CNPs was measured by 2,2-diphenyl-1-picrylhydrazy (DPPH) assay [20]. DPPH, a purple-colored, stable free radical, exhibits absorbance maxima at 517 nm. Treatment of DPPH with an antioxidant results in a change in color to yellow and a decrease in absorbance intensity. For determination of antioxidant activity, CNP suspensions of various concentrations were prepared in methanol (10–50 µg/mL, 200 µL) and to this freshly prepared DPPH, methanolic solution (0.1 Mm, 200 µL) was added so the final volume was made to 1 mL by methanol. Following 30 min of incubation, the absorbance of the solution was measured at 517 nm using methanol as the blank. The ascorbic acid solution at similar concentrations was used as standard. The antioxidant activity of CNP was determined using the following equation:(2)$$Antioxidant\;activity\;\left(\%\right)=\frac{\mathrm{DPPH}\;\mathrm{Abs}\;-\left(\mathrm{DPPH}\;\&\;\mathrm{CNP}\right)\mathrm{mix}\;\mathrm{Abs}\;}{\mathrm{DPPH}\;\mathrm{Abs}\;}$$

### 2.4. In Vivo Studies of CNPs

#### 2.4.1. Animals

The study was performed using adult female Wistar rats weighing 250–300 g, aged 6–7 months. The animals were housed in polypropylene cages (12-h light/dark cycles) under standard laboratory conditions. The room was maintained at 25 ± 2 °C and relative humidity of 60 ± 5%. Animals had free access to a commercial pellet diet, water ad libitum. All the procedures were in compliance with the Committee for the Purpose of Control and Supervision of Experiments on Animals (CPCSEA) guidelines and ethical approval (Approval date was 24 May 2019. Registration Number: 173/GO/ReBi/S2000/CPCSEA) was granted by Animal Ethics Committee of Jamia Hamdard, New Delhi.

#### 2.4.2. Treatment

Pre-training was conducted on animals and those with failure to learn were excluded. A total of 48 rats were selected and divided into 8 groups of 6 rat each: First group received i.n. saline (control group), second group received scopolamine + i.n saline (scopolamine group), third group received rivastigmine patch followed by scopolamine (rivastigmine group), 5 treatment groups received i.n. CNPs at dose of 1 mg/kg (CNP1 group), 2 mg/kg (CNP2 group), 4 mg/kg (CNP4 group), 6 mg/kg (CNP6 group), and 10 mg/kg (CNP10 group) followed by scopolamine. The rats were pretreated with either i.n. saline (10 µL) or rivastigmine patch (Exelon Patch, Novartis 13.3 mg/24 h) for 14 days, however i.n. CNPs (1, 2, 4, 6 and 10 mg/kg, 10 µL) was administered once on day 9. From day 9 onwards for next 5 days the animals (except control group) were daily injected with intraperitoneal scopolamine (1 mg/kg) to induce experimental AD. The scopolamine injection was administered 30 min post-treatment. The animals were then subjected to the cognitive behavioral test including MWM and PSA.

#### 2.4.3. Behavioral Studies

The MWM test consists of a pool (diameter 150 cm and height 50 cm) filled with water (26 ± 2 °C) up to the depth of 30 cm. The pool was hypothetically divided into four quadrants. The test was conducted in two phases, training and experimental [21]. Training trial (day 9 onwards) was conducted every day for 5 consecutive days immediately after administration of scopolamine. During training phase, an escape platform (diameter 10 cm) was kept 1 cm above the water level. Rat is placed into the pool at a random location near the edge of the pool and is allowed to locate the escape platform. Each rat was given a maximum time of 120 s to locate and climb the platform. In case the rat was unable to find the platform, it was manually guided or placed on the platform and allowed to stay there for 60 s. Total of four trials were conducted each day. The inter-trial time was two minutes and the same animal was placed into the pool at different location. Over all time required by the rat to locate and climb the platform was determined as escape latency. The experimental phase was conducted immediately after training phase and also after 24 h after the last training session. In experimental phase, the escape platform was removed from the water maze and each rat was given a maximum of 120 s to memorize the platform. The parameters such as escape latency, dwell time (time spent in the target quadrant) and path length were evaluated. A fear-based PSA test [22], to assess the ability of rats to learn from an aversive event, was conducted in two phases. Assembly used in the test consists of a well-illuminated compartment and a dark compartment interconnected with each other through an automatic sliding door. The dark chamber was equipped with an electrified grid floor. For initial 5 min, the animals were familiarized with the assembly followed by training phase after a gap of 15 min. The rat was kept in the illuminated chamber and the sliding door to the dark chamber was opened after 30 s. Step-through latency was measured as time taken by rat to enter the dark chamber. If the rat failed to enter the darkroom within 300 s, it was excluded from the test. Once the rat entered the dark chamber, the sliding door was closed and an inescapable electric foot-shock of 0.5 mA for 3 s was given through the electrified grid floor. At the end of the training trial, the rat was placed back into the cage. The assembly was cleaned thoroughly between training sessions to remove olfactory cues. An experimental trial was performed, 24-h post training phase. The time taken by the rat to completely (four paws in) enter the darkroom was recorded. During the experimental phase no foot shock was given. If the rat took more than 300 s to enter the darkroom then it was returned to the cage and step-through latency was noted as 300 s [20,22].

#### 2.4.4. Biochemical Estimation

Immediately after behavioral tests, the rats were sacrificed and the brain was removed from the skull following the heart perfusion technique to remove any blood clots that might obscure the results. The brain homogenate was prepared by washing the brain with chilled isotonic saline and homogenizing using pH 7.4 phosphate buffer. The homogenized tissue sample was cooled (4 °C) centrifuged at 10,000× *g* for 15 min and supernatant was collected. The brain homogenate so obtained was stored at −80 °C till further used [20]. For measurement of Superoxide Dismutase enzyme (SOD) activity, the brain homogenate (20 µL) was mixed with tris-HCL buffer solution prepared in ethylenediaminetetraacetic acid (pH = 8.2, 940 µL). To the mixture, pyrogallol (40 µL, 13 mM) was added and absorbance was taken at 420 nm [20]. The glutathione (GSH) level was estimated colorimetrically in the brain homogenate [23]. Briefly, 10% trichloroacetic acid was mixed with an equivalent quantity of brain homogenate. To the mixture, 5.5-dithiobisnitro benzoic acid (0.5 mL) was added. Immediately after, the absorbance was recorded at 412 nm [24].

### 2.5. Statistical Analysis

IBM SPSS Statistics version 20 as a computer software was used for Statistical analysis of the data. The experiment was done in triplicate and values were expressed as mean ± standard deviation (SD). The analysis of the result was done by one-way analysis of variance followed by Turkey posthoc test. For significance, the statistical inference level was set at (* *p* < 0.05), (** *p* < 0.01), (*** *p* < 0.001).

## 3. Results

### 3.1. Development of CNPs

The CNPs were synthesized by modified homogenous precipitation. The cerium nitrate was used at level 0.01%, 0.025% and 0.04%, methanol was used at concentration of 20%, 45% and 70%, and reaction temperarture was kept at 30 °C, 50 °C and 70 °C. A 3-factor, 3-level Box-Behnken design suggested 17 experiments. The independent variables and the responses for all the experiments are given in Table 1. The particle size was found to be smallest when concentration of cerium nitrate was 0.025 and 0.04% (Figure 1A–C). At high cerium nitrate concentration the zeta potential values were higher (Figure 1D,E). Methanol 20% and 45% resulted in large sized particles however a decrease in CNP size was observed at 70% methanol concentration. The zeta potential values were independent of methanol concentration and mostly affected by cerium nitrate concentration and temperature (Figure 1). Smallest size CNP particles with high zeta potential were observed at reaction temperature of 50 °C. Increase or decrease in reaction temperature resulted in increase in CNP particle size (Figure 1A–D). For particle size, the equation was quadratic and the F-value was significant. The predicted R square value (0.9684) was equivalent to adjusted R-square value (0.9939). The equation was:Y_1_ = +142.80 − 0.2750X_1_ − 11.90X_2_ − 7.38X_3_ − 3.22X_1_X_2_ + 2.27X_3_X_3_ + 0.3250X_2_X_3_ − 5.36X_1_^2^ + 10.74X_2_^2^ + 26.69X_3_^2^(3)

For zetapotential, the equation was quadratic and the F-value was significant. predicted R square value (0.8479) was equivalent to adjusted R-square value (0.9688). The equation was:Y_2_ = −15.36 − 6.55X_1_ + 0.1750X_2_ + 0.5250X_3_ + 0.6000X_1_X_2_ + 1.20X_1_X_3_ + 1.25X_2_X_3_ − 1.04X_1_^2^ − 0.6950X_2_
^2^ − 2.10X_3_^2^(4)

The optimized CNPs formed using 0.04% cerium nitrate, 62% methanol at reaction temperature of 50 °C has observed particle size of 134.0 ± 3.35 nm (predicted, 132.163), zeta potential of −21.8 ± 0.94 mV (predicted, −22.44) and PDI of 0.158 ± 0.0019 (Table 1, Figure 2A,B).

SEM image (Figure 2C) showed that the developed CNPs were smooth and spherical. The sphericity, nano size (105 nm) and monodispersity of the CNPs were also confirmed by TEM image, as shown in Figure 2D. The presence of Ce in CNPs was confirmed by characterstic signals in EDXA spectrum (Figure 2E). Further, broad 2θ peaks at (111), (200), (220), (311), (222), (400), and (331) in the XRD spectra (Figure 2F) indicates that the formed CNPs were nano-crystalline. Such nano-crystalline CNPs with similar XRD spectra were reported previously (25).

The formation of CNPs was confirmed by UV spectroscopy and FTIR. The UV spectra (Figure 3A) of CNPs showed two broad absorbance peak at 220 nm and 309 nm, attributed to presence of mixed oxidation states. Similar observations for CNPs was reported previously [25]. The FTIR spectra of the CNP is shown Figure 3B. Generally, the water molecules remained adsorbed over CNPs and the peaks due to the absorbed water molecules were observed in the spectra at 3358^−1^ (O-H stretching) and 1634 cm^−1^ (O-H bending). The bending vibration of C–H bonds were observed as a weak absorption band at 2107 cm^−1^ and 1463 cm^−1^. The O-C-O stretching is evident by peaks in the range of 1000–1200 cm^−1^. Further, the peak corresponding to 548 cm^−1^ in the fingerprint region, owing to O-Ce-O band confirms the presence of CNPs. Similar FTIR observations were also observed previously for CNPs [25].

### 3.2. Anti-Oxidant Activity of CNPs

The comparative antioxidant activity (as % inhibition) of CNPs (10–50 µg/mL) and ascorbic acid (standard, 10–50 µg/mL) is shown in Figure 4. Results showed that at all the concentrations, CNP demonstarted higher antioxidant activity compared to ascorbic acid. Further, the antioxidant activity of CNP at concentration of 50 µg/mL was found to be 95.40 ± 0.006%.

### 3.3. In Vivo Studies of CNPs

#### 3.3.1. Behavioral Studies

The results of the MWM test during training and experimental phase are shown in Figure 5A–C. The escape latency in the control group (89.4 ± 4.61 s) was found to be significantly lower (*p* < 0.001) than scopolamine group (129.8 ± 4.60 s), however the values were similar to rivastigmine group (89.6 ± 6.30 s). Intranasal CNP exhibited dose-dependent decrease in escape latency and the decrease in latency for CNP6 (88.4 ± 2.88 s) and CNP10 (85.4 ± 3.50 s) group was significant from scopolamine group (*p* < 0.001) but not significantly different (*p* > 0.05) from control and rivastigmine group (Figure 5A). Interestingly, the escape latency of CNP6 and CNP10 group was insignificantly different (*p* > 0.05). Similar observations were recorded for path length (Figure 5B), where values for control, rivastigmine, CNP6 and CNP10 groups were insignificantly different (*p* > 0.05). However, the values were significantly (*p* < 0.001) lower than scopolamine group. Dwell time results presented in Figure 5C showed that scopolamine group animals spend significantly less time in the target quadrant (*p* < 0.001) depicting impairment of memory. However, intranasal CNP treated rats at dose of 6 mg/kg (CNP6 group) and 10 mg/kg (CNP10 group) showed significant improvement (*p* < 0.001) in dwell time compared to scopolamine group and the values were similar to control group. Further, the results for testing and experimental phase corroborated with each other. The results of the PSA test is shown in Figure 5D. As expected, the retention latency of the scopolamine group was significantly lower (68.0 ± 2.64) than control group (190.0 ± 5.0). Further, intranasal CNP resulted in dose-dependent increase in retention latency and the retention latency of group CNP6 (188.0 ± 6.6) and CNP10 (198.0 ± 3.5) was not significantly different (*p* > 0.05) from control and rivastigmine group. Notably the results of the experimental phase and training phase were similar for both MWM and PSA test.

#### 3.3.2. Biochemical Estimation

The results of the biochemical estimation of the brain homogenate is shown in Figure 6. As shown in Figure 6A, the SOD levels of the control group (10.33 ± 0.57 µMmg^−1^) was significantly higher (*p* < 0.001) than that observed for scopolamine group (3.76 ± 0.15 µMmg^−1^) but was not significantly different (*p* > 0.05) from rivastigmine group (10.40 ± 0.30 µMmg^−1^), CNP6 group (9.86 ± 0.5 µMmg^−1^) and CNP10 group (10.66 ± 1.2 µMmg^−1^). CNP1, CNP2, CNP4 also showed increase in SOD levels but the values were significantly less (*p* < 0.001) than control group. Similar results were also observed for GSH level (Figure 6B) where scopolamine group showed significantly lower (*p* < 0.001) GSH level (1.96 ± 0.35 µMmg^−1^) compared to control (7.06 ± 0.30 µMmg^−1^). Nevertheless, the GSH levels of rivastigmine group (7.30 ± 0.45 µMmg^−1^), CNP6 group (6.93 ± 0.63 µMmg^−1^) and CNP10 group (7.46 ± 0.38 µMmg^−1^) were not significantly different (*p* > 0.05) from control.

## 4. Discussion

The link between oxidative stress and AD is well established, therefore, targeting this event might be beneficial [26]. Several antioxidants including vitamins, curcumin, berberine, quercetin, resveratrol and melatonin exhibited promising results in AD models [6]. Recently, CNPs have emerged as a regenerative antioxidants and their therapeutic potential in the neurodegenerative disorders including AD has been explored. Several in vitro studies demonstrated that CNPs decreases protein accumulation [7,8,9,10], alleviate Aβ-induced cell death [10] and trigger signal transduction pathway of neuronal survival. [7] A recent in vivo study using transgenic *Drosophila* model showed that oral CNP restore SOD levels and improved the climbing activity of elav; htau flies [8]. Current evidence suggest that intranasal delivery of drugs or particles allows their direct brain targeting [13,14,15,16,17] and we hypothesized that delivery of CNP directly into the brain would prevent the behavioral changes in experimental AD model. Intranasal delivery could also result in reduction of dose of CNP. In the present study, for the very first time we reported the efficacy of intranasal CNP in the management of cognitive effects in experimental scopolamine-induced AD.

CNPs were prepared by precipitation method following addition of ammonium hydroxide to alcoholic solution of cerium nitrate at 50 °C. The concentration of cerium nitrate, alcohol and temperature are decisive for smaller-size, stability and antioxidant property and were optimized using Box-Behnken design. Based on the responses received the optimized CNP with size 134.0 ± 3.35 nm and zeta potential −21.8 ± 4.94 was obtained at 0.04% cerium nitrate, 62% methanol and reaction temperature of 50 °C. The developed CNPs were found to be stable for 15 days at room temperature (25 ± 2 °C) and for more than 2 months in refrigerator (5 ± 3 °C). The high stability of CNPs might be due to high zeta potential value of developed CNPs [27].

SEM and TEM images revealed that the developed CNPs were spherical and uniform. The result is in agreement with previous study where ambient temperature conditions resulted in synthesis of nanosized CNPs with high antioxidant capacity [4]. The EDXA spectrum showed the presence of cerium in CNP while XRD suggested its nano-crystalline cubic and fluorite structure, as reported previously [25]. The appearance of characteristic absorbance peaks at 220 nm and 310 nm in UV-visible spectroscopy is indicative of formation of CNPs [25]. The presence of two broad absorbance peak is attributed to presence of mixed oxidation states, a property unique to CNPs [25]. Further, this ensures regular switch of cerium between two states, important for interminable and regenerative antioxidant activity of CNP [28]. Additionally, FTIR spectra showed characterstic peak of O-Ce-O band (548 cm^−1^) in finger printing region. This confirms the formation of CNPs. Cerium oxide can scavenge the free radicals which resulted in an apparent increase in the anti-oxidant activity of CNP as observed by the DPPH assay method [20].

Alzheimer is a progressive disease that destroys memory and other important mental functions and thus the efficacy of intranasal CNP in preventing behavioral changes was studied using scopolamine-induced experimental AD rat model. Scopolamine is a muscarinic cholinergic receptor antagonist and is well-accepted for developing hippocampal damaged animal models for AD [29]. Scopolamine models of AD are also utilized in several clinical trials [30]. Scopolamine causes decreases in acetylcholine level, accumulation of amyloid β protein and hyperphosphorylated tau protein and stress mediated reduction in activity of antioxidant enzymes [31]. Scopolamine induces amnesia in animals and is considered as model of choice to evaluate the protective effect of antioxidant drugs and their ability to improve cognitive functions [22]. The key feature of an AD patient is the progressive decline in cognition and behavior, mainly due to loss of neurons and synapses in the hippocampus and related areas [32]. Thus, MWM test, PSA test and brain biochemical estimation were performed. The MWM test determines hippocampal spatial memory deficits and examine age-related/AD-like deficits [22], whereas PSA test evaluates long-term contextual memory [22]. In our study, following intraperitoneal scopolamine, the experimental model was successfully induced as evident by marked impairment in cognitive ability. The escape latency, path length, dwell time and retention latency values for scopolamine group were significantly different (*p* < 0.001) from control group. In addition, scopolamine group also showed marked decrease in level of antioxidant enzymes like SOD and GSH in brain. In the present study rivastigmine transdermal patch, was used as standard for comparison and for dose-titration of i.n. CNP. Rivastigmine is a cholinesterase inhibitor approved as oral and transdermal for the treatment of AD [33]. Though, we pretreated animals with rivastigmine patch for 14 days (rivastigmine group) but i.n. CNP was administered only once (CNP1, CNP2, CNP4, CNP6 and CNP10 group). It is known that CNP can undergo redox and have the potential to regenerate [4] and thus we hypothesized that single dose might convey long-lasting effects. The results of the MWM and PSA test suggested that i.n. CNPs improve the cognitive function and the effects were dose-dependent.

The MWM results obtained for i.n. CNPs at dose of 6 mg/kg and 10 mg/kg were equivalent to control and standard treatment. Similar observations were obtained for retention latency, which increases with increase in dose of i.n. CNP and the effects were similar to control and rivastigmine group at dose of 6 mg/kg and 10 mg/kg. Interestingly, the cognitive improvement shown by 6 mg/kg i.n. CNP is not significantly different from that shown by 10 mg/kg i.n. CNP. A dose higher than 6mg/kg had conferred no appreciable additional benefits during in vivo behavioral assessment tests. Thus, our finding suggest that 6 mg/kg is dose of choice of i.n. CNP for protection against behavioral changes during progression of AD, however studies involving large number of animals is needed for conclusion. Further, results of biochemical estimation showed that the levels of SOD and GSH are preserved following pre-treatment with single dose of i.n. CNPs. This could be due to uptake of nanosized CNPs in brain following i.n. administration. The high brain concentrations of drug and nanoparticles following intranasal delivery is well documented [13,14,15,16,17].

In conclusion, intranasal CNPs as a single dose of 6 mg/kg exhibited significant improvement of cognitive functions (spatial and non-spatial memory) in the scopolamine-induced model of AD. Thus, intranasal CNPs, with strong and regenerative antioxidant activity, could be a prospective therapeutic for the management of AD associated symptoms.

## Figures and Tables

**Figure 1 pharmaceutics-14-00756-f001:**
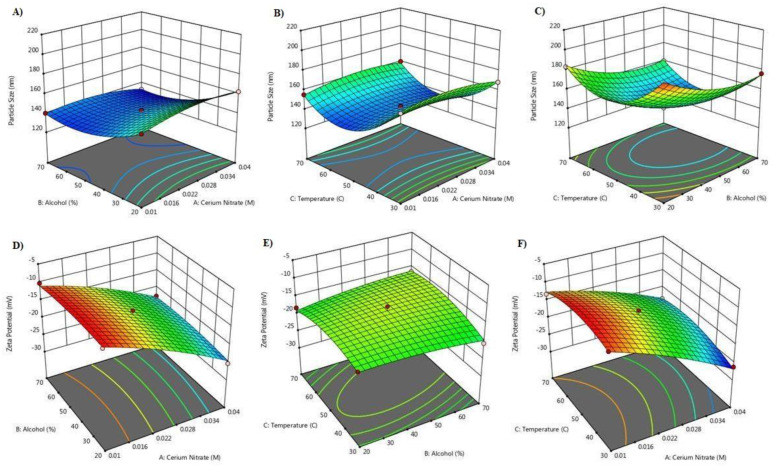
3D-response surface plots depicting the impact of cerium nitrate concentration, methanol and reaction temperature on CNP particle size and zeta potential. (**A**–**C**): Shows effect of independent variables on particle size, (**D**–**F**): Shows effect of independent variable on zeta potential.

**Figure 2 pharmaceutics-14-00756-f002:**
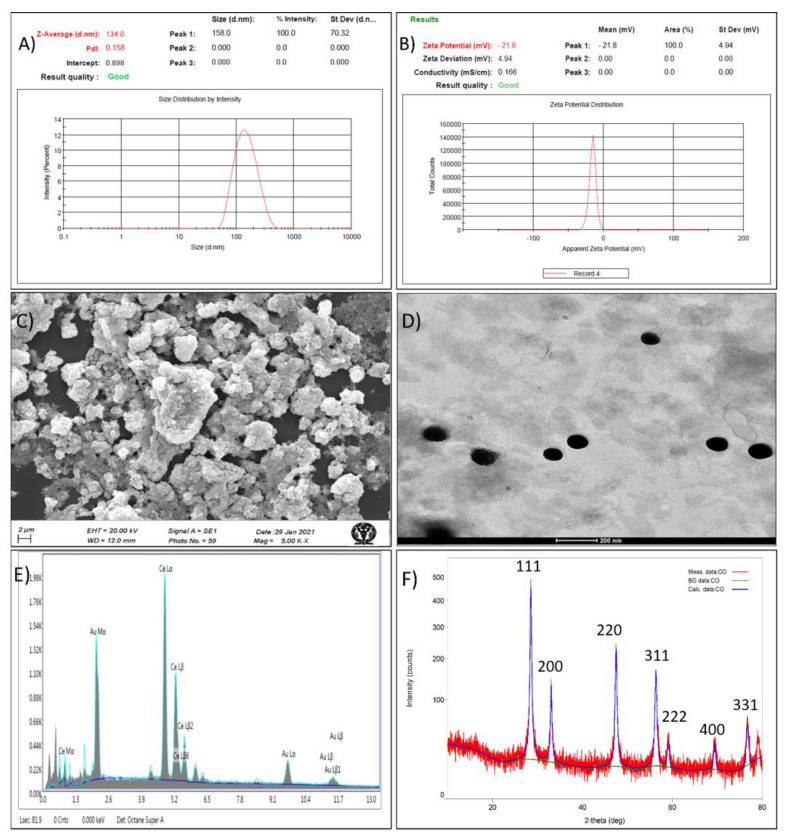
Characterization of optimized cerium oxide nanoparticles, (**A**) Particle size, (**B**) Zeta potential, (**C**) SEM image, (**D**) TEM image, (**E**) EDXA spectra, and (**F**) XRD.

**Figure 3 pharmaceutics-14-00756-f003:**
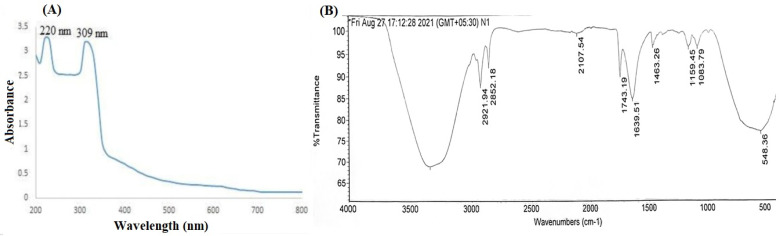
Structural conformation of CNPs, (**A**) UV spectra (**B**) FTIR spectra.

**Figure 4 pharmaceutics-14-00756-f004:**
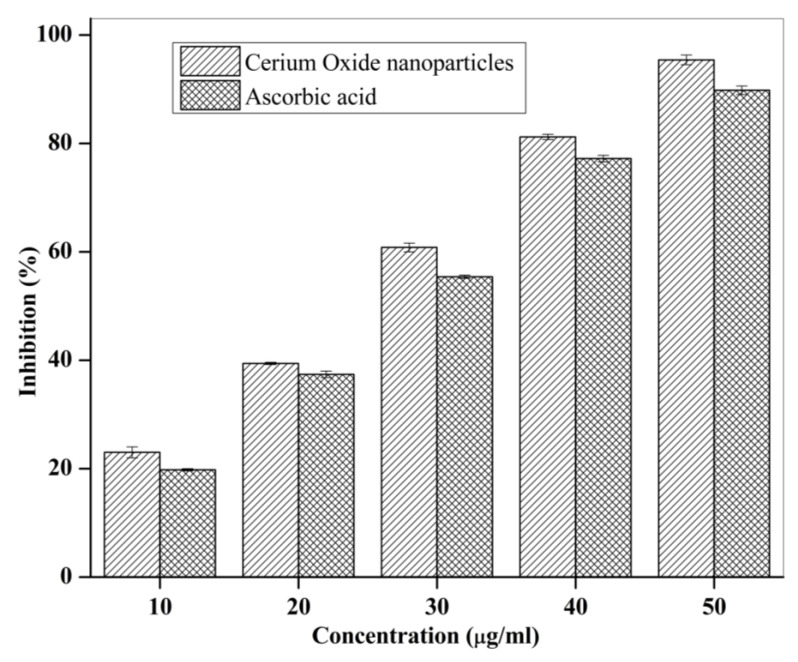
In vitro antioxidant activity of CNPs and ascorbic acid (standard).

**Figure 5 pharmaceutics-14-00756-f005:**
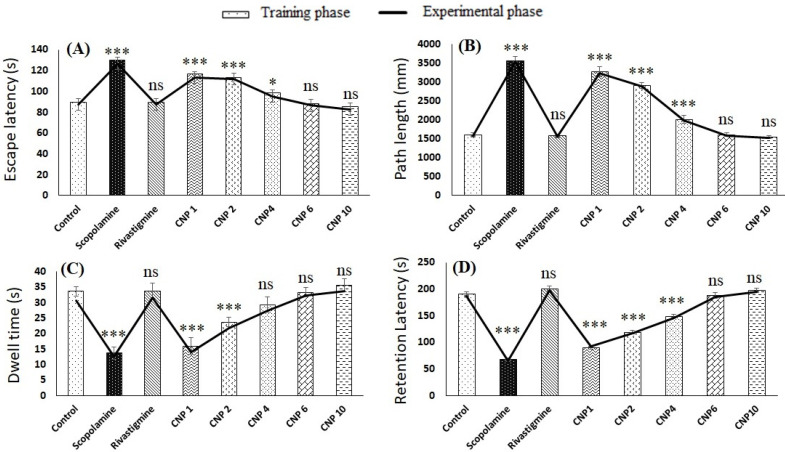
Effect of CNP on cognitive function (**A**) MWM escape latency, (**B**) MWM path length, (**C**) MWM dwell time and (**D**) PSA retention latency. (* *p* < 0.05), (*** *p* < 0.001), (ns = non significant).

**Figure 6 pharmaceutics-14-00756-f006:**
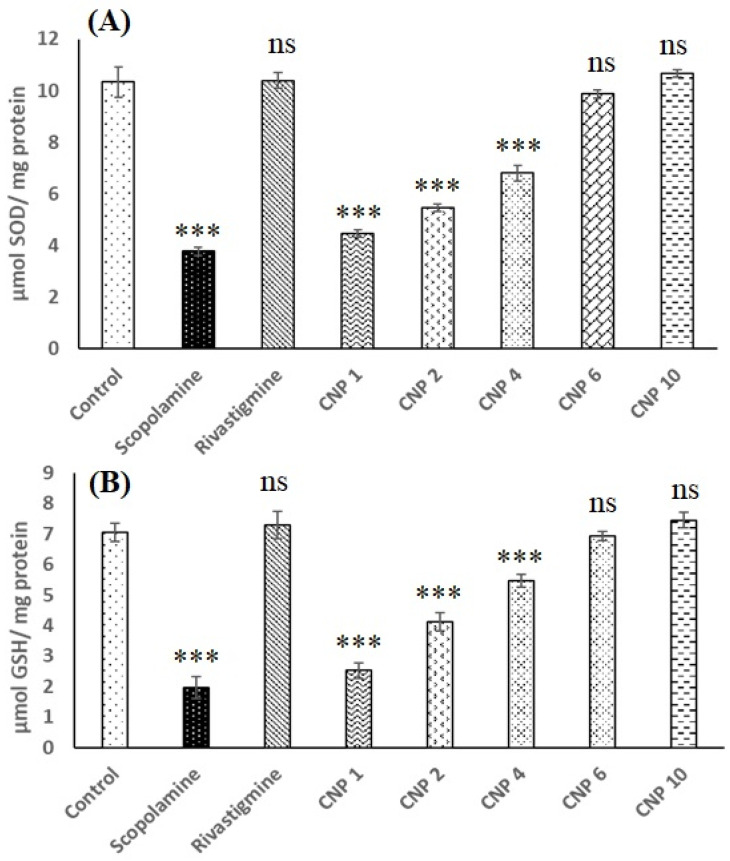
Effect of CNP on (**A**) SOD level and (**B**) GSH level in scopolamine induced female Wistar rats. (*** *p* < 0.001, ns = non-significant).

**Table 1 pharmaceutics-14-00756-t001:** Box-Behnken design with independent variables, experimental runs and responses (*n* = 3).

Formulation Code		Independent Variables		Dependent Variables	
	X_1_	X_2_	X_3_	Y_1_	Y_2_
CNP1	0.01	70	50	140.2 ± 3.50	−10.2 ± 0.25
CNP2	0.01	45	70	155.4 ± 4.19	−13.2 ± 0.35
CNP3	0.04	70	50	132.4 ± 3.31	−22.6 ± 0.56
CNP4	0.025	70	70	160.2 ± 4.80	−16.4 ± 0.49
CNP5	0.025	20	30	200.9 ± 6.42	−17.4 ± 0.55
CNP6	0.025	20	70	183.4 ± 6.23	−18.2 ± 0.61
CNP7	0.025	45	50	143.6 ± 4.73	−15.8 ± 0.39
CNP8	0.025	45	50	143.1 ± 4.86	−15.4 ± 0.40
CNP9	0.04	45	70	160.2 ± 6.08	−23.4 ± 0.72
CNP10	0.025	70	30	176.4 ± 4.93	−20.6 ± 0.57
CNP11	0.025	45	50	141.1 ± 4.37	−15.6 ± 0.46
CNP12	0.01	20	50	157.5 ± 3.93	−10.4 ± 0.26
CNP13	0.04	20	50	162.6 ± 4.87	−25.2 ± 0.80
CNP14	0.025	45	50	142.8 ± 5.71	−15.8 ± 0.42
CNP15	0.025	45	50	143.4 ± 3.87	−14.2 ± 0.39
CNP16	0.01	45	30	172.6 ± 4.83	−11.2 ± 0.32
CNP17	0.04	45	30	168.3 ± 5.38	−26.2 ± 0.65

X_1_ = Cerium nitrate (M), X_2_ = Methanol conc (%), X_3_ = Temperature (°C), Y_1_ = CNP size (nm), Y_2_ = Zeta potential (mV).

## Data Availability

This study did not report any data (All the data is included in the current manuscript).

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
