# Peer review of "Intranasal Cerium Oxide Nanoparticles Ameliorate Cognitive Function in Rats with Alzheimer’s via Anti-Oxidative Pathway"

_pharmaceutics, 2022, doi:10.3390/pharmaceutics14040756_

Round 1

Reviewer 1 Report

This manuscript by Danish et al reports an intranasal delivery strategy of cerium oxide nanoparticles which can ameliorate cognitive function in Alzheimer rat model. Overall, the experiments were well designed and the data can support their conclusion. The authors also provide detailed experimental procedures so that others can easily repeat their results. Thus, I recommend the manuscript for publication after minor revision.

  1. In Figure 2F, the reviewer suggests the author assign each of these peaks with the proper crystal phase (e.g. (110)).
  2. In Figure 3 and 4, the reviewer suggests the author to add the group information in the Figure caption. If possible, please the original data points in each of these figures.
  3. In the Introduction section, when the author mentioned “In the case of intranasal delivery, BBB is bypassed as the nanoparticles are taken up by olfactory cells of the nasal mucosa, reach the olfactory bulb in the limbic system, and are distributed to a different region of the brain by the olfactory and trigeminal nerve.” The author may also need to cite the reference which shows the same merit of this claim: NMR in Biomedicine, 2013, 26, 1176-1185.   “It is well reported that nanoparticles within the size range 200 nm can easily target the brain” Reference: The Enzymes, 2018, 43, 31-65; Acta Pharmaceutica Sinica B, 2021, 11, 3447-3464.

Reviewer 2 Report

Overall:

The paper entitled “Intranasal cerium oxide nanoparticles ameliorates cognitive function in alzheimer rat model via anti-oxidative pathway” focuses on neuroprotective capacity of cerium oxide nanoparticles against scopolamine-induced rat model of Alzheimer’s disease (AD). The Authors conducted behavioral studies which showed that intranasal cerium oxide nanoparticles exhibited similar neuroprotective effects to rivastigmine during passive avoidance and Morris water maize tests. Also, the antioxidant activity of cerium oxide nanoparticles resulted in up-regulation of GSH and SOD level in brain homogenates after the treatment. In such a prestigious, high IF journal, I would suggest extended biochemical and molecular analyses. The publication (in the current version) shows very promising but only preliminary results.

Therefore, at the stage, I recommend reject this article from Pharmaceutics.

Comments and Suggestions for Authors:

  1. The Authors did not comment why did they use scopolamine-induced rat model of Alzheimer’s disease. What are the advantages and disadvantages of the model?
  2. Why the Authors used only female rats?
  3. The Authors did not provide the BBB permeability of the cerium oxide nanoparticles.
  4. Is it a good idea to perform biochemical tests (determination of GSH and SOD) after behavioral testing? Does this not interfere with basic level?
  5. Please provide the information about the sensitivity and selectivity of the methods used to determine GHS and SOD level.
  6. There is no information about the origin of rats in groups and between groups. Whether the rats were related / were from the same litters.
  7. The Figures, in particular Fig. 3 has poor quality.
  8. Presenting results as A, B, C, D, E, F, G, H on the X axis is difficult to interpret. I would suggest adding a legend in the charts or a detailed description of the X-axis.
  9. Some bars lack standard deviations (e.g., Fig. 4A – bar A).
  10. I suggest standardizing the appearance of the figures, some bars are colored, others have patterns.

Editorial comments:

  1. The manuscript has been uploaded in incorrect format. The Authors did not use the MDPI form.
  2. I suggest the English editing services.

Reviewer 3 Report

The authors developed cerium oxide nanoparticles (NPs) for the intranasal delivery for Alzheimer’s disease (AD) antioxidant therapy. The theme is interesting and timely, since finding new therapy strategies for AD is urgent. The economic and social burden of this neurodegenerative disease is substantial, reflecting in high health care costs as well as loss of productivity due to morbidity and premature death. Drug delivery to the brain is very challenging and hinders the success of many therapeutic drugs. Nose-to-brain delivery is a promising strategy and is currently extensively studied due to the ability to bypass the blood-brain barrier and to increase drug bioavailability in the brain. Though, it is not clear the novelty of this work, since cerium oxide NPs for AD were already reported in the literature (DOI’s: 10.2174/157341309788185523; 10.1007/s13205-021-02706-x; 10.1038/cdd.2014.72). The main novelty of this work should be presented clearly in the introduction. The developed nanoformulations were physiochemically characterized using adequate methodologies, and there in vivo efficacy was studied in rats.

Though, some questions should be addressed before publication. Below the authors can find some suggestions and questions.

1) Keywords: please avoid repeating words from the title, such as Cerium Oxide nanoparticles

Abstract:

 2) the authors stated: “Nanoceria (NC) was synthesized using homogenous precipitation method and optimized through Design expert software”. It is more important to give details of the type of the used experimental design used (Box–Behnken design) instead of the used software.

3) please add mean value ± standard deviation for size, PDI and zeta potential of the optimized formulation

4) please include full name of the used methodologies (e.g. EDXA, XDR)

5) Regarding the maximum percentage inhibition obtained in DPPH experiments, please uniformize the standard figures. Also, the % symbol is missing.

6) Section 2.3 (Box Behnken design):  In table 1, do the authors intend to say that no constrains were applied for zeta potential, or that the goal was to obtain neutral NPs? Please clarify. I suppose it is no constrains, because in table 4,  we can see that the optimized formulation has negative zeta potential. Why did the authors study zeta potential as a response variable, if they have no intention on optimize it. Also, positively charged NPs are associated with longer retention time in the nasal cavity, and therefore are more adequate for intranasal delivery.

7) Section 2.4 Characterization of Cerium Oxide Nanoparticles: Antioxidant experiments should be in a separated section and not included in section 2.4

8) Table 2: please indicate in table caption the number of replicas for each formulation (n)

Results:

9) In the section “Effect of independent factors on particle size and zeta potential”: I suggest removing from the main text F-value was 292.57, predicted R and adjusted R-squared, since these are presented in table 3.

10) table 4: please standardize significant figures

11) Antioxidant activity by DPPH assay results: please standardize significant figures

12) Figure 3A: the authors do not present the standard deviation. Did the authors only perform one experiment? If so, experiments should be repeated in triplicates to show the reproducibility of the method and mean + SD should be given.

The manuscript needs extensive English editing.

Round 2

Reviewer 2 Report

The authors responded to all my comments/questions. After taking into account the comments suggested by the entire group of reviewers, the publication became more valuable which will be appreciated by readers. In my opinion, the paper is now ready for publication (some editorial changes will be needed).